# The Superiority of TiO_2_ Supported on Nickel Foam over Ni-Doped TiO_2_ in the Photothermal Decomposition of Acetaldehyde

**DOI:** 10.3390/ma16155241

**Published:** 2023-07-26

**Authors:** Beata Tryba, Piotr Miądlicki, Piotr Rychtowski, Maciej Trzeciak, Rafał Jan Wróbel

**Affiliations:** Department of Catalytic and Sorbent Materials Engineering, Faculty of Chemical Technology and Engineering, West Pomeranian University of Technology in Szczecin, Pułaskiego 10, 70-322 Szczecin, Poland; piotr.miadlicki@zut.edu.pl (P.M.); piotr.rychtowski@zut.edu.pl (P.R.); tm44864@zut.edu.pl (M.T.); rafal.wrobel@zut.edu.pl (R.J.W.)

**Keywords:** thermo-photocatalysis, nickel foam, Ni-doped TiO_2_, acetaldehyde decomposition

## Abstract

Acetaldehyde decomposition was performed under heating at a temperature range of 25–125 °C and UV irradiation on TiO_2_ doped by metallic Ni powder and TiO_2_ supported on nickel foam. The process was carried out in a high-temperature reaction chamber, “The Praying Mantis^TM^”, with simultaneous in situ FTIR measurements and UV irradiation. Ni powder was added to TiO_2_ in the quantity of 0.5 to 5.0 wt%. The photothermal measurements of acetaldehyde decomposition indicated that the highest yield of acetaldehyde conversion on TiO_2_ and UV irradiation was obtained at 75 °C. The doping of nickel to TiO_2_ did not increase its photocatalytic activity. Contrary to that, the application of nickel foam as a support for TiO_2_ appeared to be highly advantageous because it increased the decomposition of acetaldehyde from 31 to 52% at 25 °C, and then to 85% at 100 °C in comparison with TiO_2_ itself. At the same time, the mineralization of acetaldehyde to CO_2_ doubled in the presence of nickel foam. However, oxidized nickel foam used as support for TiO_2_ was detrimental. Most likely, different mechanisms of electron transfer between Ni–TiO_2_ and NiO-TiO_2_ occurred. The application of nickel foam greatly enhanced the separation of free carriers in TiO_2_. As a consequence, high yields from the photocatalytic reactions were obtained.

## 1. Introduction

Photothermal catalysis combines photochemical and thermochemical contributions of sunlight and has emerged as a rapidly growing and exciting new field of research [1,2,3]. Photothermal processes can enhance the yield of photo-Fenton reactions [4,5].

Photothermal catalysis with the application of semiconductors doped by metallic nanoparticles allows for more effective harvesting of the solar spectrum, including low-energy visible and infrared photons.

Certain metallic nanoparticles (NPs), such as Au, Ag, Cu, and others, exhibit unique optical properties related to the localized surface plasmon resonance (LSPR), which can be tuned via varying their size and shape across the entire visible spectrum [6,7]. 

In the LSPR process, hot charge carriers are generated, which possess higher energies than those induced by direct photoexcitation. These energetic carriers can be transferred to the adsorbate surface or the conductive band of semiconductors or can relax internally and dissipate their energy by local heating of the surroundings, causing a thermal effect on the material. This photothermal effect has been extensively applied in a large number of fields, including cancer therapy, the degradation of pollutants, CO_2_ reduction, hydrogen production through water splitting, and other chemical reactions such as ex. catalytic steam reforming or the hydrogenation of olefins [2,8,9,10,11,12,13,14,15]. Photothermal effects can also be obtained in non-plasmonic structures through direct intraband and/or interband electronic transitions. For instance, Sarina et al. [16] demonstrated that non-plasmonic metal NPs supported on ZrO_2_ could catalyze cross-coupling reactions at low temperatures under visible light. According to the authors, upon irradiation with UV light, electrons could shift to high-energy levels through interband transitions, and only those with enough energy could transfer to the LUMO of the adsorbed molecules, just like in the case of plasmonic metal NPs. When excited with low-energy visible-IR light, the electrons were not energetic enough to be injected into adsorbate states, thus contributing to the enhancement of the reaction rate by means of thermal effects. Altogether, photochemical and thermal effects contribute to photothermal performance in non-plasmonic semiconductors.

The size, shape, and quantity of the plasmonic nanoparticles contributing to the Photothermal effect are important. The temperature increase is proportional to the square of the nanoparticle radius. However, it has been theorized that the temperature increment due to the irradiation of a single nanoparticle becomes negligible, but the light-induced heating effect could be strongly enhanced in the presence of a large number of NPs due to collective effects [2]. Theoretical calculations have demonstrated that larger plasmonic NPs provide a larger number of hot carriers, but they display energies close to the Fermi level since most of the absorbed energy is dissipated either by scattering or by heating. In contrast, smaller NPs exhibit higher energies but they show very short lifetimes [2]. Lower-sized plasmonic NPs show blue shift absorption edge due to the quantum size effect [17]. All these parameters are extremely important during the design of nanomaterial for thermophotocatalytic processes.

The most intensively explored plasmonic NPs used in photothermal processes so far are based on Au, Ag, Pd, Cu, and Pt nanoparticles; less information is reported on other metals such as Ni or Co. The thermophotocatalytic decomposition of acetaldehyde was reported for TiO_2_ doped by Pt, Ag, Pd, or Au nanoparticles [12,18,19,20,21]. Of all these metals doped to TiO_2_, Pt had the highest activity due to its strong thermocatalytic effect at elevated temperatures. The activity of metal or metal oxide doped to TiO_2_ depended on the oxidation state of the metal and its adsorption abilities towards acetaldehyde species. The other structures, such as metal nanowires coated by TiO_2_, have also been prepared and tested for acetaldehyde removal [22,23]. There are some reports on the application of nickel foam with loaded TiO_2_ for different purposes, such as anodes for lithium-ion batteries [24], hydrogen evolution from water splitting [25], toluene and acetaldehyde decomposition [26,27], nitrogen oxides removal [28], and others. Nickel foam has also been used as a support and catalyst for different materials in photocatalytic processes [29,30,31]. The main advantage of the application of nickel foam as a composite with other photocatalyst is the significant improvement in charge separation in the photocatalytic material.

Although some reports on the application of nickel foam with loaded TiO_2_ have already been published, in this study, we propose for the first time the mechanism of the thermophotocatalytic decomposition of acetaldehyde under UV light in the presence of TiO_2_ supported on nickel foam, and this is compared with others which occur when oxidized nickel foam or nanosized Ni particles added to TiO_2_ are utilized. 

## 2. Materials and Methods

The TiO_2_ of anatase phase was obtained by a two-step process: hydrothermal treatment of amorphous titania (Police Chemical Factory, Gliwice, Poland) in deionized water at 150 °C under pressure of 7.4 bar for 1 h, followed by annealing in a pipe furnace at 400 °C under Ar flow for 2 h. The obtained material consisted of anatase (97 wt%) with an average crystalline size of 15 nm and small quantity of rutile (3 wt%). The measured BET surface area of this material was 167 m^2^/g. Our previous work presented the detailed physicochemical properties of this material [32].

Nickel foam (Jiujiang, China) with purity of 99.8% had the following parameters: thickness, 1.5 mml; porosity, 95–97%; surface density, 300 g/m^2^. Nickel powder (Warchem, Poland) had a purity of 99.8%. 

The other chemicals used were as follows: p-benzoquinone (HPLC purity, >99.5%, Fluka Analytical, Darmstadt, Germany), terephthalic acid (TA) (purity 98%, Sigma-Aldrich, Saint Louis, MO, USA) ethylenediaminetetraacetic acid (EDTA) (Pure Chemical Standards—Elemental Microanalysis), AgNO_3_ (Polish Chemical Factory, Gliwice, Poland).

XRD measurements of nickel foams were performed using the diffractometer (PANanalytical, Almelo, The Netherlands) with a Cu X-ray source, λ = 0.154439 nm. Measurements were conducted in the 2θ range of 10–100° with a step size of 0.013. The applied voltage was 35 kV, and the current was 30 mA.

SEM/EDS images were obtained using a field emission scanning electron microscope with high resolution (UHR FE-SEM, Hitachi, Japan).

Oxidation of nickel foams was carried out in a muffle furnace (Czylok SM-2002, Jastrzębie-Zdrój, Poland) at 500 °C for 1 h.

The chemical composition was determined through X-ray photoelectron spectroscopy (XPS) analysis. The measurements were conducted using a commercial multipurpose ultra-high vacuum (UHV) surface analysis system (PREVAC, Rogow, Poland). A nonmonochromatic XPS source and a kinetic electron energy analyzer (SES 2002; Scienta, Taunusstein, Germany) was used. The spectrometer was calibrated using the Ag 3d5/2 transition. The XPS analysis utilized Mg Kα (h = 1253.6 eV) radiation as the excitation source. 

The size of nickel powdered nanoparticles was determined using an atomic force microscope (AFM; NanoScope V Multimode 8, Bruker, Billerica, MA, USA) with a silicon nitride probe in ScanAsyst mode. Prior to measurement, the samples were dispersed in isopropyl alcohol and then drop-casted onto a silicon wafer. Surface topography images were obtained using NanoScope Analysis software (v1.40r1), whereas particles sizes were evaluated via ImageJ software (2023).

Thermophotocatalytic decomposition tests of acetaldehyde were conducted using a high=temperature reaction chamber (Harrick, Pleasantville, NY, USA), as shown in Figure 1. During the test, continuous FTIR measurements were conducted using Thermo Nicolet iS50 FTIR instrument (Thermo, Waltham, MA, USA). 

UV irradiation was conducted through a quartz window, with an illuminator equipped with a fiber optics and a UV-LED 365 nm diode with an optical power of 415 mW (Warszawa, Poland, LABIS). The intensity of incident UV radiation was measured by a photoradiometer, HD2102.1 (TEST-THERM, Kraków, Poland). The obtained value of UV intensity measured at the surface of reactor cover window equaled 20 W/m^2^. Gases were supplied from bottles through two inlets using mass flow meters: the first one was acetaldehyde in nitrogen, 300 ppm (Messer, Police, Poland); the second one was a synthetic oxygen with purity 5.0 (Messer). Gases have been mixed in proportions to create a synthetic air composition. After the thermophotocatalytic process, the gas stream was directed to the gas chromatograph (GC) equipped with an automatically dosing sample loop (GC-FID, SRI, Menlo Park, CA, USA). Acetaldehyde concentration was determined from recorded chromatograms in GC. Outlet gas stream from GC was flowing through the CO_2_ sensor (APFinder CO_2_, ATUT Company, Kraków, Poland) in order to monitor the quantity of CO_2_ formed upon acetaldehyde mineralization. The contact time of acetaldehyde species with the photocatalytic material was varied through the application of different flow rates of a gas stream, from 15 to 30 mL/min. Then, the optimal flow rate was selected for all the photocatalytic tests, taking into account the highest quantity of acetaldehyde removal over the irradiated TiO_2_.

The catalytic systems that have been used in the tests are as follows: (a) TiO_2_; (b) a thin layer (1 mm) of TiO_2_ supported on KBr which is inert to acetaldehyde; (c) a thin layer of TiO_2_ supported on Ni foam; (d) a thin layer of TiO_2_ supported on the oxidized Ni foam; (e) TiO_2_ blended with the nickel powder in the various amount, 0.5–5 wt%. In Figure 2, the scheme of the materials compositions used for the photocatalytic tests of acetaldehyde decomposition is illustrated.

To determine the dominant species which take part in the photocatalytic reactions, some measurements were performed in the presence of oxygen radicals and holes scavengers. For that purpose, p-benzoquinone, ethylenediaminetetraacetic acid (EDTA), and terephthalic acids were used as scavengers for O_2_^−•^, h^+^, and ^•^OH species, respectively. For the quenching reactions, 0.1 g of TiO_2_ photocatalyst was mixed with 0.01 g of each scavenger separately, and then such mixture was loaded on the purified nickel foam. Excessive amounts of scavenger reagents were utilized to guarantee the complete capture of related radicals, similar as in the other experiments reported by Q. Zeng et al. [20]. Acetaldehyde decomposition in the presence of TiO_2_ and scavenger was carried out in the high-temperature chamber at 50 °C under UV irradiation. As a blank test, the mixture of 0.1 g TiO_2_ with 0.01 g of KBr was used because KBr was revealed to be chemically inert for acetaldehyde gas.

## 3. Results

Preliminary acetaldehyde decomposition was carried out at various flow rates of a gas stream through the reactor, from 15 to 30 mL/min. The diameter of the sample holder equaled 0.6 cm^2^. For lower flow rate of a gas, the contact time of acetaldehyde with TiO_2_ surface was longer, and the percentage of removed acetaldehyde was higher. However, by increasing the flow rate of a gas, higher loading of TiO_2_ with acetaldehyde species was obtained. There is a certain optimum amount of pollutant which can be decomposed on TiO_2_ surface at given time. In Figure 3, the dependence of acetaldehyde decomposition over time via the quantity of acetaldehyde loading on the titania surface is illustrated.

These measurements revealed that the maximal quantity of acetaldehyde decomposition could be obtained when the surface loading with these species was 17 ppm per min·cm^2^ (for a flow rate of a gas equaled 20 mL/min). For lower flow rate used, such as 15 mL/min, there is higher percentage of acetaldehyde removal from a gas stream; however, the maximum yield of the photocatalytic system has not yet been reached. Therefore, for other photocatalytic tests, the flow rate of a gas stream equal to 20 mL/min was applied.

The results from the thermophotocatalytic decomposition of acetaldehyde over TiO_2_ and UV-LED irradiation are illustrated in Figure 4. 

The performed measurements indicated that the conversion of acetaldehyde on TiO_2_ could be enhanced at elevated temperatures. In the case of a thick layer of TiO_2_ being used (without KBr) (Figure 4b) the highest acetaldehyde conversion was observed at 125 °C; however, this process was not stable and indicated a gradual falling-off in degradation rate, reaching its maximum after 180 min of UV irradiation with a conversion of 50%. Contrary to that, the application of a thin layer TiO_2_ supported on KBr allowed us stabilize the process, but the highest conversion of acetaldehyde was lower than in case of using TiO_2_ only and reached 47% at 75 °C. In Table 1, the results from the measurements of CO_2_ in the outlet stream of reacted gases are presented.

These results indicated that mineralization of acetaldehyde was higher in the case of the photocatalytic system used with a thin layer of TiO_2_. Acetaldehyde can be oxidized over TiO_2_ in the dark in the presence of oxygen; however, its complete mineralization occurs in the presence of reactive radicals. Most likely, TiO_2_ at the bottom of reactor chamber was not activated by UV light and formed products of acetaldehyde decomposition at the lower part of TiO_2_ were not mineralized, but acted as scavengers for reactive radicals formed at the surface of TiO_2_ on the top of reactor. Therefore, for higher quantities of TiO_2_ used, acetaldehyde conversion was higher, but its mineralization to CO_2_ dropped down.

Total mineralization of acetaldehyde can be summarized via the following equation:CH_3_CHO + 5/2O_2_ → 2CO_2_ + 2H_2_O. (1)

From the above equation, it is deduced that from one mole of acetaldehyde, two moles of carbon dioxide are formed. The highest quantity of CO_2_ (330 ppm) was formed in the conditions of a thin layer of TiO_2_ and a temperature of 100 °C, where removal of acetaldehyde was determined to be 41%. In the case of total mineralization of acetaldehyde, the quantity of CO_2_ should be around 197 ppm, but it was 330, which attributes to selectivity much more than 100%. The reason for such an unusual phenomenon is that at the same time acetaldehyde and the adsorbed products of its degradation were mineralized, acetaldehyde was adsorbed onto the TiO_2_ surface and transformed to other formate and acetate species; hence, other acetaldehyde molecules can be adsorbed on the titania surface after total decomposition of the byproducts. Therefore, the obtained values of CO_2_ concentration did not refer to the percentage decrease in acetaldehyde concentration at a given time only.

In Figure 5, the results from the thermophotocatalytic decomposition of acetaldehyde over TiO_2_ supported on the nickel foam are presented.

High decomposition of acetaldehyde (around 85%) was observed in the presence of TiO_2_ supported on Ni foam and in reaction temperatures of 100–125 °C. The slight decrease in acetaldehyde decomposition over time is due to by-products blocking the catalyst’s active centers. However, the efficiency of this process significantly decreased in the presence of oxidized Ni foam; the decomposition of acetaldehyde dropped down to 33%. It has to be mentioned that Ni foam as received was much less active than that after the photocatalytic process. Therefore, all the results presented in Figure 5a were obtained for reused nickel foam.

In Table 2, the results from measurements of CO_2_ after the thermophotocatalytic process of acetaldehyde decomposition conducted for TiO_2_ supported on reused Ni foam and oxidized at 500 °C are presented.

Application of Ni foam with TiO_2_ not only enhanced the decomposition rate of acetaldehyde but also doubled its mineralization degree. The highest formation of CO_2_ was noted at 100 °C (472 ppm) and this, again, was higher than total mineralization of acetaldehyde, which was calculated to be 408 ppm for 85% of its decomposition at given time.

In order to explain the mechanism of acetaldehyde decomposition over TiO_2_ supported on nickel foam, some photocatalytic experiments were performed with the addition of scavengers for some active species. In Figure 6, the results of the measurements are presented.

Addition of p-benzoquinone (p-BQ) to TiO_2_, which acted as a scavenger of superoxide anion radicals, greatly suppressed acetaldehyde decomposition. Contrary to that, the addition of terephthalic acid (TA) to TiO_2_ (hydroxyl radicals scavenger) enhanced acetaldehyde decomposition in comparison with a blank test performed for TiO_2_ without the addition of any scavengers. In the presence of EDTA (hole scavenger), acetaldehyde decomposition decreased but to a lesser extent than in case of p-BQ addition. These experiments showed that superoxide anion radicals played the dominant role in the removal of acetaldehyde. Similar results were obtained by other researchers [20].

In Figure 7 the results from the photocatalytic decomposition of acetaldehyde over TiO_2_ blended with nanosized Ni powder are presented.

The quantity of doped Ni slightly affected the efficiency of acetaldehyde decomposition, which ranged from 38 to 43%. At a temperature of 125 °C, the process yield was the highest and reached around 50% for 1% of doped Ni to TiO_2_. Although the acetaldehyde decomposition rate on TiO_2_ doped with Ni was quite comparable with that for TiO_2_, the mineralization degree was low; the maximum value of formed CO_2_ was 150 ppm, which was reached for 1% doped Ni at a reaction temperature of 75 °C; and for the other tests, the quantities of formed CO_2_ were lower.

AFM analyses of surface topography with loaded Ni particles indicated that their mean size was c.a. 30 nm, as shown in Figure 8.

In Figure 9, XRD patterns of Ni foam as received and after oxidation at 500 °C in air are illustrated.

After oxidation of Ni foam at a temperature of 500 °C, new reflexes emerged which were assigned to the NiO phase; however, their intensities were much lower than those related to Ni.

In Figure 10 the SEM images of nickel foam are shown at different magnifications.

The performed SEM images showed some impurities on the surface of nickel foam as received from the manufacturer (Figure 10a). After the photocatalytic process, these impurities disappeared (Figure 10b). They were removed through the reactive oxygen species formed on TiO_2_ upon UV irradiation. This was confirmed by the other experiment, in which TiO_2_ supported on the commercial nickel foam was used with flowing air without acetaldehyde but under UV irradiation. The same effect of nickel foam purification was observed.

The oxidized nickel foam showed a corrugated surface on the grain fringes, indicating the proceeding oxidation process.

In Figure 11, mapping of the oxidized nickel foam is depicted. The green area indicates the oxygen distribution. It is clearly seen that strongly corrugated surface is covered with elemental oxygen.

In the next step, XPS analyses were performed for nickel foam as received (fresh) after the photocatalytic process and that oxidized at 500 °C in air. Recorded XPS spectra for C1s, Ni2p_3/2_, and O1s signals are presented in Figure 12.

The elemental surface content from XPS analyses is presented in Table 3.

It should be noted that this is an average content of elements from a depth of c.a. 1 nm with an assumption that the elements are homogeneously distributed, which is not valid. Therefore, these values should be taken as an approximation. The nickel foam oxidized at 500 °C exhibits no carbon over the surface. The other nickel foams exhibit a significant amount of carbon, which attenuates the Ni2p signal. This attenuation explains the low values of Ni2p signals.

The deconvolution of C1s signal was performed on the basis of [33]. The carbon species were observed on non-oxidized nickel foam before and after the photocatalytic process. The prominent shoulder on the left-hand side of the signal is related to the carbon–oxygen species present over the surface of carbon. The intensity of C1s signal increased after the process, i.e., the carbon coverage has increased. However, both samples showed similar composition to an oxygen carbon species.

XPS spectra of Ni 2p signals indicated that the highest intensity of Ni species had the sample oxidized at 500 °C. This can be attributed to the absence of carbon over the surface of this sample. In the case of other nickel foams, the intensity of the Ni2p signal was lower due to the carbon coverage, which attenuates the signal of Ni2p. This signal can be deconvoluted to Ni^0^, Ni^2+^, Ni^3+^ (852.3, 853.8, and 855.8 eV, respectively [28]), and satellite signals. The signal of Ni^0^ is negligible. The metallic Ni is present in the nickel foam, but the XPS sampling depth is c.a. 1 nm. Therefore, the surface covered with oxides and carbon attenuates the signal from beneath. Nickel foam oxidized at 500 °C clearly showed signals related to Ni^2+^ and Ni^3+^, whereas the other samples revealed mostly signal of Ni^3+^.

The O1s can be unambiguously deconvoluted only for the sample oxidized at 500 °C due to the absence of carbon. In the case of other samples, there are carbon–oxygen species which contribute to O1s signal. This renders very difficult the unambiguous deconvolution of the O1s spectrum. In the case of the sample oxidized at 500 °C, two distinct components can be observed: one can be attributed to bulk nickel oxide, and the other to the -OH groups present over the surface of nickel oxide. It is assumed that a thin layer of Ni(oxy)hydroxide was formed on the surface of the oxidized nickel foam.

## 4. Discussion

Nickel foam used as a support for TiO_2_ showed spectacular properties for enhancing the photocatalytic decomposition of acetaldehyde under UV light irradiation. However, the doping of the nanosized Ni particles to TiO_2_ did not bring any advantages in either acetaldehyde decomposition or mineralization. Partly oxidized nickel foam also deteriorated the rate of acetaldehyde removal from a gas stream, although some other researchers indicated the superiority of the oxidized nickel foam over that non-oxidized [27]. These different properties of nickel materials could be explained by various reaction mechanisms that occurred in the presence of TiO_2_. 

The performed photocatalytic tests of acetaldehyde decomposition in the presence of some reactive scavengers species indicated that superoxide anion radicals are the dominant species contributing to acetaldehyde decomposition. These superoxide anion radicals can be formed through the electron capture by the adsorbed oxygen. It is stated that these species are greatly produced at the interfacial border between TiO_2_ and nickel foam, where nickel foam is the electron supplier. Therefore, in the presence of nickel foam, acetaldehyde mineralization was greatly enhanced. Mineralized byproducts of acetaldehyde decomposition released space for the adsorption of another acetaldehyde species; therefore, the total efficiency of acetaldehyde removal was increased.

In the case of oxidized Ni foam, there is high probability of electron injection from the conductive band of NiO through the electron transfer zone—Ni (oxy)hydroxide to the TiO_2_ due to the p–n heterojunction mechanism between these two semiconductors [34]. XPS measurements showed that there is a thin layer of Ni (oxy)hydroxide formed on the surface of oxidized Ni foam. There are some reports that this layer can facilitate an interfacial electron transfer [35]. On the other hand, holes from the titania valence band can migrate to the lower energy valence band of NiO or can form hydroxyl radicals with hydroxyl groups present on the interfacial border between the oxidized nickel foam and TiO_2_. The performed photocatalytic tests with hydroxyl radicals scavenger (TA) indicated that acetaldehyde decomposition increases when there is a quenching of hydroxyl radicals. Therefore, hydroxylated TiO_2_ can be disadvantageous for acetaldehyde decomposition. Most likely, the formation of hydroxyl radicals through the reaction of hydroxyl groups with holes is competitive with the reaction of acetaldehyde oxidation:h^+^ + ^−^OH → ^•^OH(2)
h^+^ + 3CH_3_CHO + O_2_ → 2CH_3_COOH + CH_3_CO^•^ + H^+^.(3)

However, hydroxyl groups take part in the oxidation of byproducts formed upon acetaldehyde decomposition, such as ex. acetic acid:CH_3_COOH + ^•^OH → CO_2_ + H_2_O + CH_3_^•^.(4)

Nevertheless, hydroxyl radicals can be also formed in a double-electron oxygen reduction pathway [36]:O_2_ + 2H^+^ + 2e^−^ → H_2_O_2_(5)
H_2_O_2_ + e^−^ →^•^OH + HO^−^.(6)

The other route of hydroxyl radicals formation can be through single-electron reduction conducted to form superoxide anion radicals, which, in further steps, yield in H_2_O_2_ production:e^−^ + O_2_ → O_2_^−•^(7)
O_2_^−•^ + H^+^ → HO_2_^•^(8)
2 HO_2_^•^ → H_2_O_2_ + O_2_.(9)

Therefore, it is stated that the formation of superoxide anion radicals is greatly demanded, together with the formation of hydroxyl radicals through the reaction pathways (5–9). An excess of hydroxyl radicals could recombine with electrons; hence, some reactive species were then extinguished and both acetaldehyde mineralization and decomposition decreased. Some tests of AgNO_3_ reduction by electrons formed upon excitation of TiO_2_ supported on nickel foam (oxidized and not) were performed. The results were placed in the Appendix A. These experiments showed a slightly lower reduction of AgNO_3_ species in the presence of oxidized nickel foam (NiO). It means, that a lower quantity of electrons were formed in the presence of TiO_2_/NiO, so non-oxidized nickel foam (Ni) improved the separation of charge carriers in TiO_2_. Other researchers also observed improved charge separation in cases of application of nickel foam with other photocatalyst [30,36]. Contrary to that, in the presence of oxidized nickel foam, recombination of charge carriers could take place.

Doping of nanosized Ni powder to TiO_2_ did not bring any spectacular effect in boosting the photocatalytic process of acetaldehyde decomposition. Nanosized nickel after excitation can generate electrons, which can be transferred to the conductive band of TiO_2_ or to the adsorbed species on its surface. On the other hand, the lifetime of these electrons is very short, and they can undergo back transfer with heat generation. 

It was proved that acetaldehyde decomposition could be enhanced at an elevated temperature. It is assumed that in the increased temperature, higher quantity of O_2_^−•^ and H_2_O_2_ species are formed due to the increased mobility of electrons in the Ni-conductive foam. Both H_2_O_2_ species and the hydroxyl radicals formed upon their reduction can take part in the oxidation of acetaldehyde and its conversion products. Therefore, in the presence of Ni foam, the mineralization degree of acetaldehyde was greatly enhanced.

The other researchers reported increased decomposition of acetaldehyde on TiO_2_ doped with 0.5% of Pt due to the spillover of oxygen from Pt to TiO_2_ surface, which could oxidize byproducts of acetaldehyde conversion [19].

The other situation was noted in case of Au-doped TiO_2_, where Au particles served as the active adsorption centers for acetaldehyde [37]. In that case, Au nanoparticles oxidized acetaldehyde to acetic acid; then, dissociated ions of acetate were transferred to the TiO_2_ surface, where they underwent photochemical decomposition. 

However, these studies showed that the doping of nanosized Ni species to TiO_2_ did not enhance its photocatalytic activity, even at an elevated temperature. The possible direct transfer of electrons from dopant to TiO_2_ could be detrimental to this process. Reduction of TiO_2_ can increase its hydrophilicity. More advantageous is transfer of electrons from nickel foam to the adsorbed oxygen species.

The mechanism of electron transfer between nickel material and TiO_2_ was important in obtaining an increased efficiency of acetaldehyde decomposition. In this case, nickel foam with some adsorbed oxygen species on its surface was suitable because of the enhanced separation of free carriers and increased acetaldehyde mineralization through its contribution to the formation of active radicals. 

## 5. Conclusions

The performed studies on the thermophotocatalytic decomposition of acetaldehyde showed superior properties of nickel foam used as a support for TiO_2_. It was evidenced that nickel foam could improve the photocatalytic properties of TiO_2_ for acetaldehyde conversion at room temperature form 31 to 52%, and from 40 to 85% at 100 °C, doubling the mineralization degree. Even at lower temperatures, such as 50 °C, the conversion of acetaldehyde on TiO_2_ supported on nickel foam was high, reaching 80% (for TiO_2_ itself, maximal conversion of acetaldehyde was obtained at 75 °C with value of 46%). These results indicate the synergistic effect between nickel foam and TiO_2_, which can also be utilized in other photocatalytic reactions because charge separation in TiO_2_ is a key factor responsible for its photocatalytic activity. Using nickel foam as a support for TiO_2_ gives much space for the interaction of species between the interface boarder; therefore, high enhancement of the photocatalytic yield was observed. The preparation of photocatalytic composites based on nickel foam creates a new direction in materials development.

## Figures and Tables

**Figure 1 materials-16-05241-f001:**
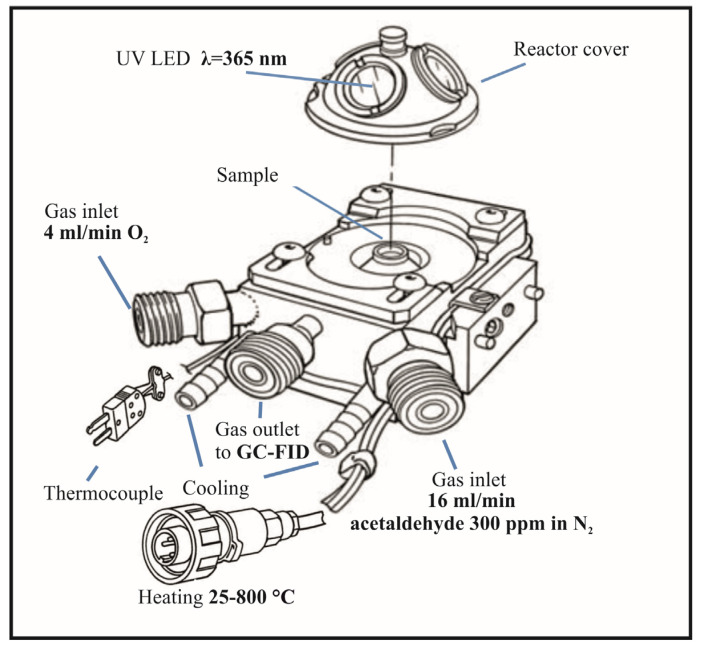
The scheme of the Praying Mantis™ high-temperature reaction chamber.

**Figure 2 materials-16-05241-f002:**
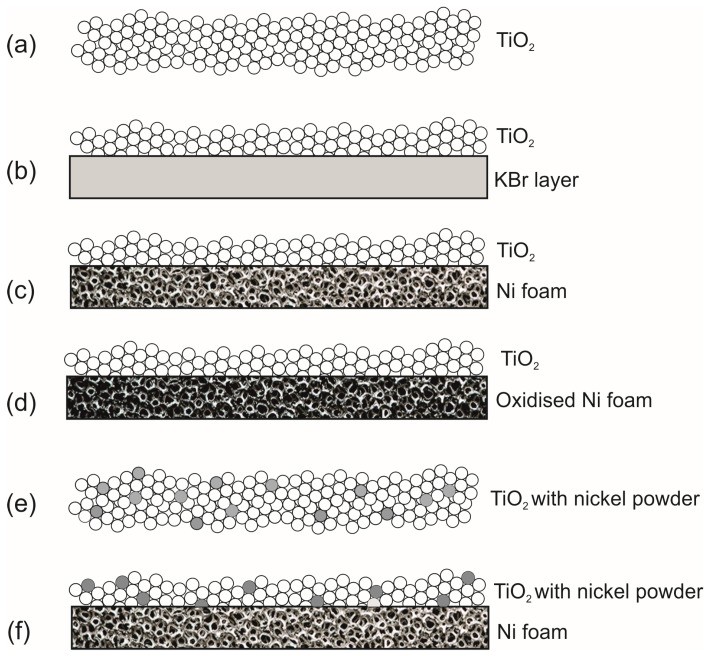
Catalytic systems used during thermophotocatalytic decomposition of acetaldehyde.

**Figure 3 materials-16-05241-f003:**
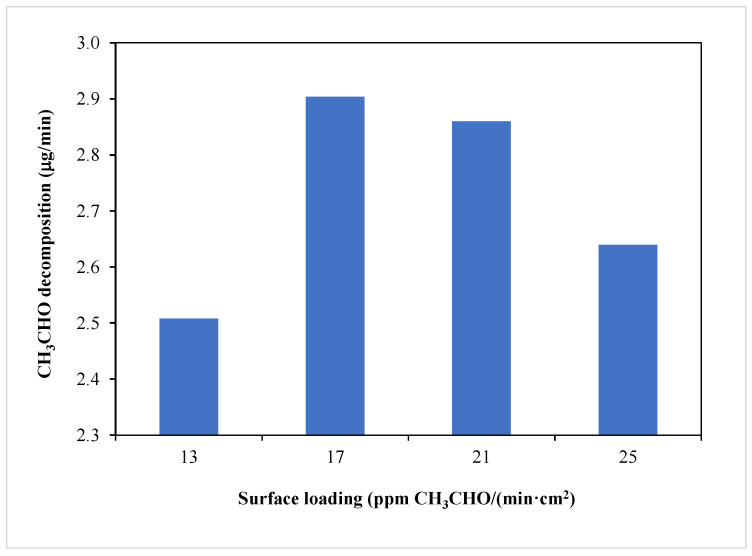
Dependence of acetaldehyde decomposition on its loading on the titania surface at a given time.

**Figure 4 materials-16-05241-f004:**
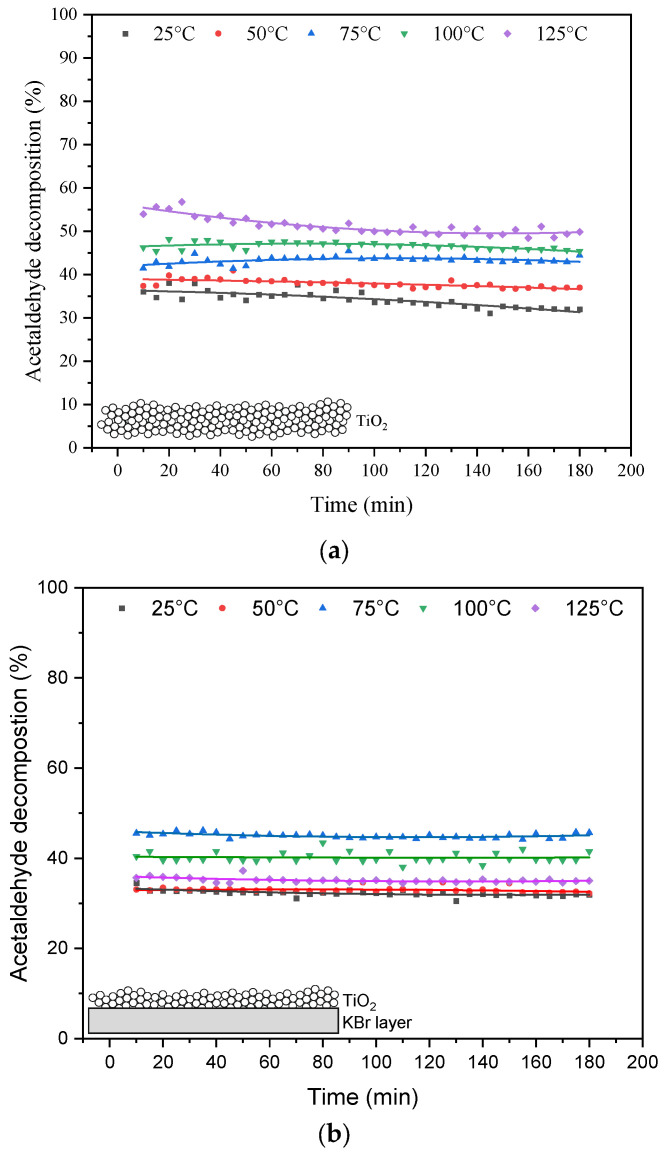
Photocatalytic decomposition of acetaldehyde under UV irradiation at various reaction temperatures in the presence of (**a**) TiO_2_; (**b**) TiO_2_ supported on KBr.

**Figure 5 materials-16-05241-f005:**
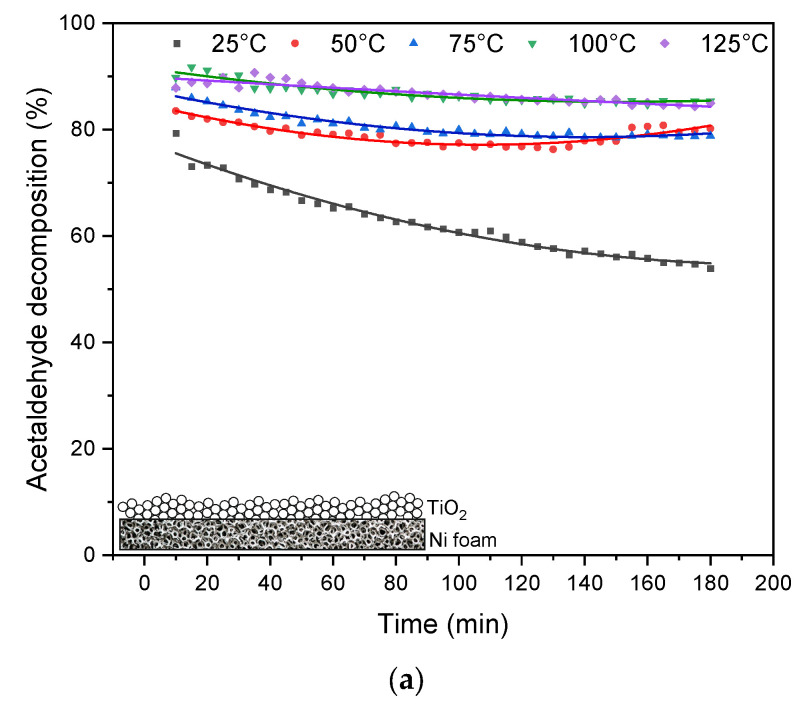
Photocatalytic decomposition of acetaldehyde under UV irradiation at various reaction temperatures in the presence of (**a**) TiO_2_ supported on Ni foam; (**b**) TiO_2_ supported on oxidized Ni foam.

**Figure 6 materials-16-05241-f006:**
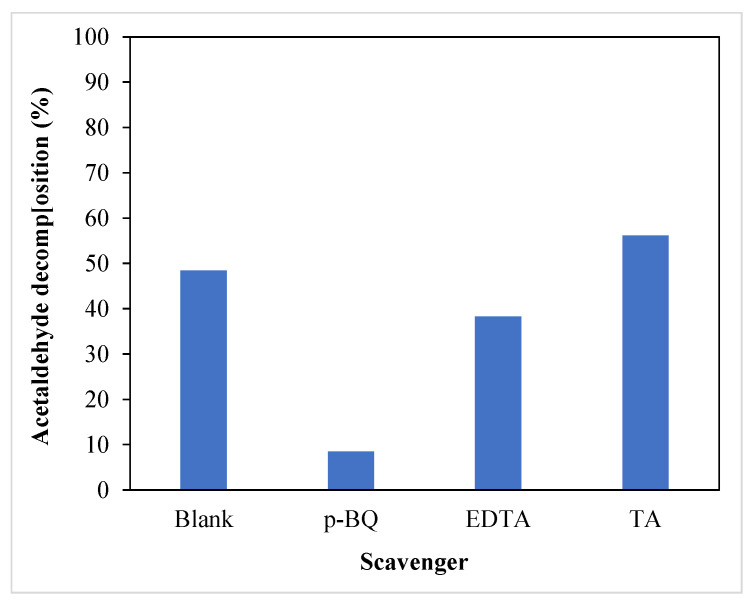
Photocatalytic decomposition of acetaldehyde in the presence of various scavengers.

**Figure 7 materials-16-05241-f007:**
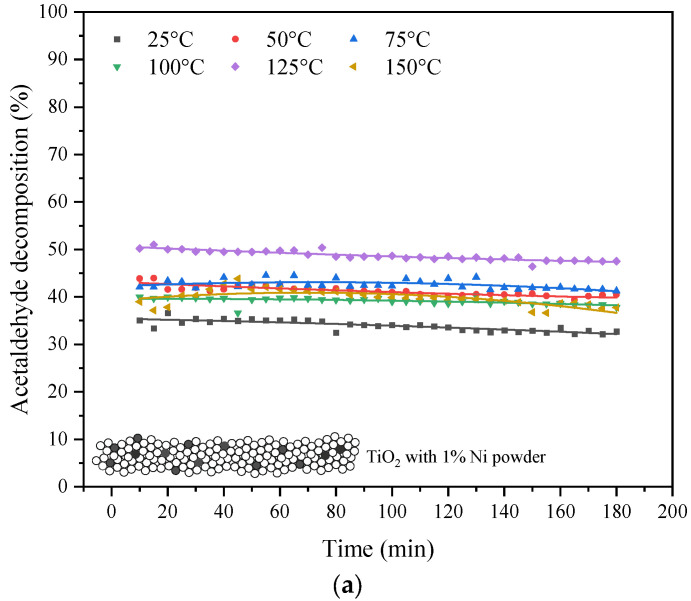
Photocatalytic decomposition of acetaldehyde under UV irradiation in the presence of TiO_2_ doped with Ni powder: (**a**) in various reaction temperatures with 1% of doped Ni; (**b**) for different quantities of doped Ni with a reaction temperature of 100 °C.

**Figure 8 materials-16-05241-f008:**
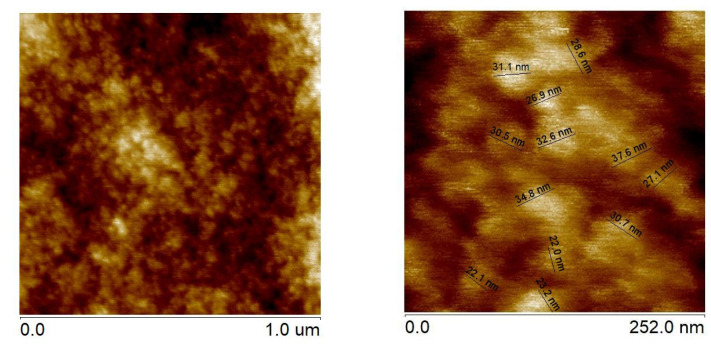
Surface topography of nickel nanoparticles.

**Figure 9 materials-16-05241-f009:**
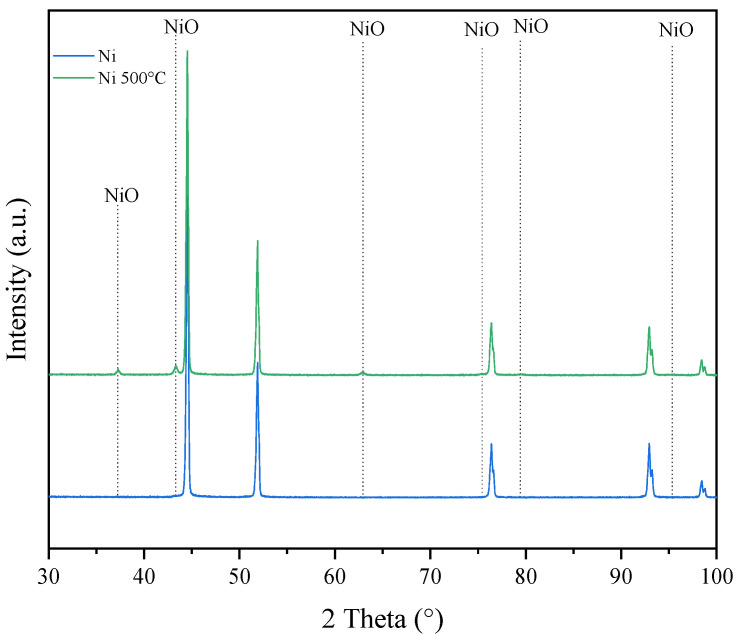
XRD patterns of Ni foam before and after oxidation at 500 °C in air.

**Figure 10 materials-16-05241-f010:**
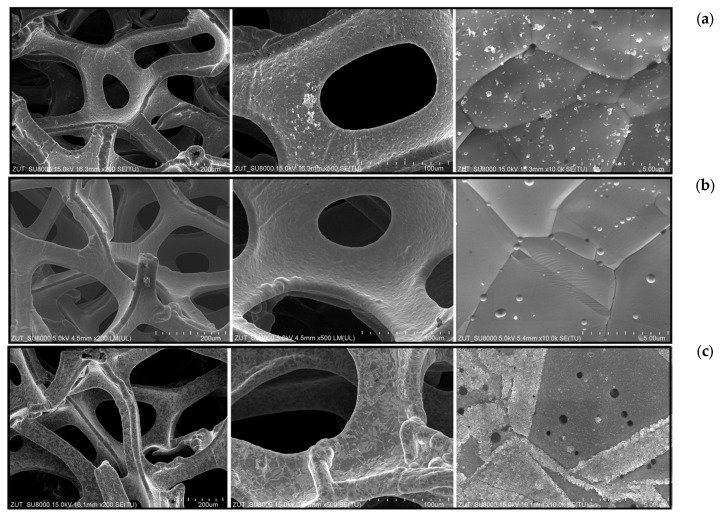
SEM images of a nickel foam: (**a**) as received; (**b**) after photocatalytic process; and (**c**) oxidized at 500 °C in air.

**Figure 11 materials-16-05241-f011:**
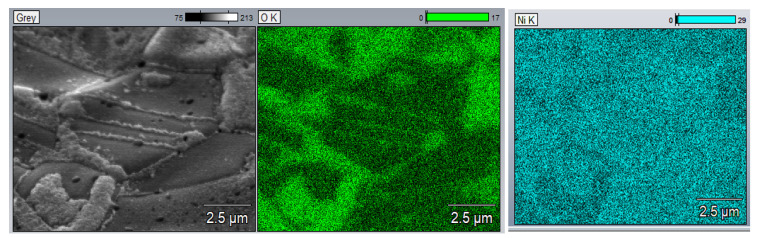
Elemental mapping of nickel foam surface oxidized at 500 °C in air.

**Figure 12 materials-16-05241-f012:**
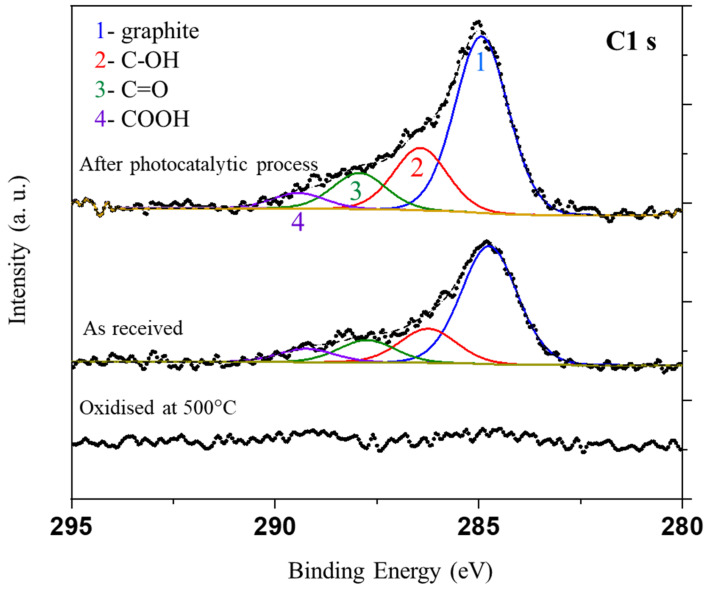
XPS spectra of nickel foams for C1s, Ni2p_3/2_, and O1s signals.

**Table 1 materials-16-05241-t001:** Content of CO_2_ in a gas stream after acetaldehyde decomposition on TiO_2_.

Reaction Temperature (°C)	CO_2_ (ppm)	
	TiO_2_	TiO_2_ on KBr
25	220	245
50	235	245
75	232	300
100	210	330
125	225	290

**Table 2 materials-16-05241-t002:** Content of CO_2_ in a gas stream after acetaldehyde decomposition on TiO_2_.

Reaction Temperature (°C)	CO_2_ (ppm)	
	TiO_2_ on Reused Ni Foam	TiO_2_ on Ni Foam Oxidized at 500 °C
25	364	-
50	409	<100
75	438	<100
100	472	<100
125	423	-

**Table 3 materials-16-05241-t003:** Elemental analysis of nickel foam surface.

Nickel Foam	Elemental Surface Content (% at.)		
	O1s	Ni2p	C1s
As received	55	9	36
After photocatalysis	42	6	53
Oxidized at 500 °C	62	38	-

## Data Availability

Data will be available in repozytorium ZUT.

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
