# Peer review of "The Superiority of TiO2 Supported on Nickel Foam over Ni-Doped TiO2 in the Photothermal Decomposition of Acetaldehyde"

_materials, 2023, doi:10.3390/ma16155241_

Round 1
Reviewer 1 Report
Comments,
The manuscript fabricated a TiO2 supported on the nickel foam over Ni- TiO2 to decompose acetaldehyde. It was evidenced, that nickel foam could improve photocatalytic properties of TiO2 for acetaldehyde conversion at room temperature. In summary, the work is interesting and could be considered for publication in Materials if the following questions are addressed.
(1) What are the advantages of nickel foam as a titanium dioxide carrier?
(2) The photothermal effect of Ni nanoparticles on formaldehyde degradation needs further analysis. Some references may be helpful for this study (Journal of Colloid and Interface Science, 2022, 622, 52a6-537; Applied Catalysis B: Environmental, 2021, 292, 120198).
(3) In the process of photocatalytic degradation of formaldehyde, the dominant active species should be confirmed.
(4) The charge transfer between NiO and TiO2 should be further investigated by some characterization.
Author Response
- What are the advantages of nickel foam as a titanium dioxide carrier?
The nickel foam improves separation of charge carriers and is an electron source. Performed photocatalytic tests of acetaldehyde decomposition at the presence of some reactive species scavengers indicated, that superoxide anion radicals are dominant species contributed in acetaldehyde decomposition. These superoxide anion radicals are formed through the electron capture by the adsorbed oxygen. It is stated, that these species are greatly produced at the interfacial border between TiO2 and nickel foam, where nickel foam is the electron supplier. Therefore at the presence of nickel foam acetaldehyde mineralisation was greatly enhanced. Mineralised byproducts of acetaldehyde decomposition released space for adsorption of another acetaldehyde species and therefore the total efficiency of acetaldehyde removal was increased.
- The photothermal effect of Ni nanoparticles on formaldehyde degradation needs further analysis. Some references may be helpful for this study (Journal of Colloid and Interface Science, 2022, 622, 52a6-537; Applied Catalysis B: Environmental, 2021, 292, 120198).
These references were added to the manuscript and were discussed in the text.
- In the process of photocatalytic degradation of formaldehyde, the dominant active species should be confirmed.
Some photocatalytic tests of acetaldehyde decomposition at the presence of scavengers for various active species were performed and added to the manuscript. These measurements allowed to determine the dominant active species contributed in acetaldehyde decomposition.
- The charge transfer between NiO and TiO2should be further investigated by some characterization.
We have performed some test of electron trapping by AgNO3 species loaded on TiO2/nickel foam for both, oxidised and not oxidised nickel foam. For that purpose the samples were irradiated under UV light and the changes of the chemical surface structure after irradiation were monitored by UV-Vis spectroscopy. AgNO3 species at the presence of TiO2 and UV can be reduced to the metallic Ag nanoparticles, which are easy analysed by UV-Vis absorption technique. The intensity of absorption peak, characteristic for Ag nanoparticles was related to the quantity of electrons participated in AgNO3 reduction. Discussion of the obtained results was added to the text.
Reviewer 2 Report
This paper shows the different performances of nickel-TiO2 photocatalysts in acetaldehyde photothermal oxidation. The interest is the comparison with various arrangements of combining the two species.
Some issues should be solved.
The treated flowrate is very low, and the measures could be affected by mass transfer limitations.
Tests at different contact times should be performed
The evaluation of activation energy can be performed and can highlight this effect.
The selectivity to CO2 must be calculated.
The specific surface area of the samples should be introduced.
The intensity of radiation should be mentioned
Author Response
- The treated flowrate is very low, and the measures could be affected by mass transfer limitations.
Tests at different contact times should be performed.
Some additional tests of acetaldehyde decomposition for various contact time were performed. The results were added to the manuscript and were discussed in the text.
- The evaluation of activation energy can be performed and can highlight this effect.
There was not possible to determine activation energy in this type of experiment, because process was carried out with continuous flow of gas without recirculation, so the change in substrate concentration was stable in time.
- The selectivity to CO2 must be calculated.
Selectivity to CO2 was calculated, however in some cases calculated selectivity exceeded 100%, because at the same time acetaldehyde and adsorbed products of its degradation were mineralised. Acetaldehyde adsorbs on TiO2 surface and is going transformation to other formate and acetate species, so another acetaldehyde molecules can be adsorbed on titania surface after total decomposition of byproducts. Therefore the obtained values of CO2 concentration did not refer to the percentage decrease of acetaldehyde concentration at given time only. The explanation of this phenomenon was included in the text.
- The specific surface area of the samples should be introduced.
Information about the specific surface area of samples was added to the manuscript.
- The intensity of radiation should be mentioned.
The radiation intensity was measured and added to the text.
Reviewer 3 Report
I have reviewed your manuscript “Superiority of TiO2 supported on the nickel foam over Ni 2 doped TiO2 in the photo-thermal decomposition of acetaldehyde” submitted to" materials" Journal, (Manuscript number: materials-2490280).
Its performed studies on thermo-photocatalytic decomposition of acetaldehyde used a nickel as a support for TiO2 and compared with other types of prepared catalysts (TiO2, TiO2 supported on KBr, and TiO2 supported on oxidized Ni foam.).
However, in my opinion, the manuscript is still descriptive, urgently requiring an improvement of discussion of the state-of-the-art and experimental data, based on the most relevant and available literature. My specific comments are reported below:
The issues of other previous studies are missing in the introduction
Check the figures and modified in the correct form its mixed between a) and a) inside the figures
In the study the effect of concentration of the acetaldehyde in ratio are missing.
The correlation between the characterization of the materials and their catalytic activity still not clear its needs to improve.
As you mentioned in the manuscript “It has to be mentioned, that Ni foam as received was much less active than that after photocatalytic process” this part need to give a clear explanation.
please check the manuscript and correct the English mistakes.
Author Response
The issues of other previous studies are missing in the introduction
Introduction was updated, some other published papers were mentioned.
Check the figures and modified in the correct form its mixed between a) and a) inside the figures
Figures were checked and corrected.
In the study the effect of concentration of the acetaldehyde in ratio are missing.
We have performed different flow rate of acetaldehyde gas through the reactor, so concentration of acetaldehyde in time was various, these results were added to the text and were discussed.
The correlation between the characterization of the materials and their catalytic activity still not clear its needs to improve.
Correlation between characterization of materials and their catalytic activity was highlighted in the discussion part. The other experiments of acetaldehyde decomposition with addition of some active species scavengers were performed to clearly explain mechanism of acetaldehyde decomposition on different materials used. Discussion was improved.
As you mentioned in the manuscript “It has to be mentioned, that Ni foam as received was much less active than that after photocatalytic process” this part need to give a clear explanation.
The other test was performed, in which Ni foam was loaded with TiO2 and irradiated by UV light with simultaneous flowing of air. After this process the nickel foam was activated, what was confirmed by using it for acetaldehyde decomposition. So, oxygen reactive radicals formed on TiO2 upon UV irradiation could remove some impurities. Explanation of this was inserted to the text of manuscript.
Round 2
Reviewer 1 Report
The authors replied all the comments, and the revised manuscript is suitable for publication.
Author Response
Thank you
Reviewer 2 Report
The suggestion to evaluate the activation energy of the photocatalyzed reaction was not satisfied.
The experiment with scavengers should be previously described in the methods.
the language could be revised
Author Response
To evaluate the activation energy, another photocatalytic system should be used, than that, which is presented in this paper, because we have used photoreactor with continuous flow of a gas stream, to determine activation energy we should change system onto reactor with circulated gas stream, therefore we can not do it as the reviewer wished. We will be thinking about it in the future experiments.
Experiment with scavengers was described in the section Methods.
English was corrected.
Reviewer 3 Report
Dear authors,
After carefully I reviewed the modifications and corrections you made on your revision manuscript. The changes you implemented have significantly improved the content, clarity, and overall quality of the draft. In my opinion, the manuscript no longer requires any further modifications.
Author Response
Thank you.